# Optimizable LLM Planning: A Branch-and-Bound Framework for Complex Tasks

## Abstract

Optimizing reasoning and action planning with LLMs under limited computational budgets is a fundamental problem in solving complex tasks that require multi-step decomposition. Existing approaches, including greedy step-by-step reasoning and tree-based search, remain largely budget-blind. For these methods, budget is enforced by ad-hoc stopping rules rather than treated as explicit optimization objectives, which prevents them from adapting the depth and breadth of reasoning to different levels of budget. To address this, we introduce **O**ptimizable **LLM P**lanning (OLP), a branch-and-bound framework that formulates planning as a budgeted optimization problem for task success. At each step, each candidate plan is expanded by decomposing the task into an immediately solvable subtask and a residual subtask. For the residual, the planner estimates lower and upper bounds on utility calibrated from reward and cost signals, where the reward model is adaptable to different execution operators (e.g., retrieval, LLM reasoning). This calibration enforces budget feasibility and supports principled ranking of candidate plans. The bound-guided search avoids unrolling entire trajectories, focuses exploration on candidates whose upper bounds dominate, and prunes branches whose upper bounds fall below competing lower bounds, enabling effective exploration of both depth and breadth under budget constraints. We instantiate this general framework for retrieval-augmented generation (RAG) problems that require reasoning. Across multiple benchmarks, our framework achieves higher accuracy than strong agentic baselines using different search algorithms while substantially reducing computation, demonstrating the effectiveness of making planning explicitly optimizable under budget constraints.

## 1 Introduction

ReAct-style planning (Yao et al., 2023b), which alternates between reasoning and action steps, has become widely adopted in LLM-powered systems for solving complex tasks. However, at each step, ReAct planning makes a greedy decision about the next action, making it inherently *non-optimizable* under varying constraints such as computational budgets. While budget information could in principle be provided to the planner, mere awareness of the budget does not enable ReAct to optimize its plan based on the budget constraints. Once an action is chosen, the trajectory becomes fixed, and the system cannot globally adjust its reasoning path in response to the remaining budget. Tree-based search algorithms (Yao et al., 2023a; Hao et al., 2023), such as Monte Carlo Tree Search (MCTS) (Świechowski et al., 2023), broaden the search space by introducing exploration through selection, expansion, simulation, and backpropagation. Although MCTS allows a fixed simulation budget (e.g., number of rollouts or time), this budget is used only as an external cap on the number of simulations. The exploration process remains heuristic-driven and does not provide a formulation that allows steps along the trajectories to be explicitly optimized with respect to budget. In practice, search is still governed by ad-hoc limits, such as maximum steps or rollouts (Muennighoff et al., 2025), rather than dynamically scaling with available resources.

This limitation becomes clear in realistic tasks. Consider the question *"(In 2023) How has Apple's net sales in Greater China changed over time relative to total net sales?"* with EDGAR[1] as the

---

[1]EDGAR is a financial database provided by the US SEC.

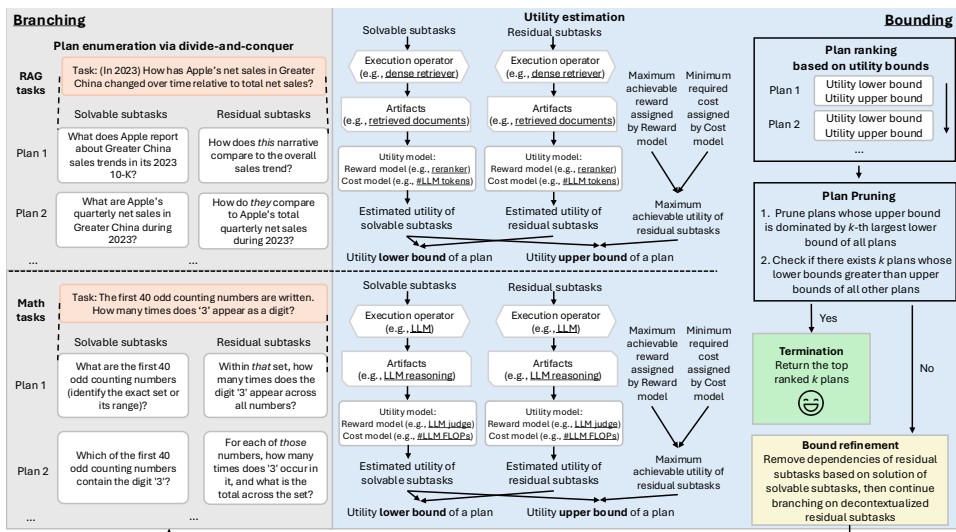

Figure 1: An overview of our branch-and-bound LLM planning framework, OLP, instantiated on complex tasks such as RAG and Math, each with suitable execution operators, reward models, and cost models (underlined). The framework generates alternative task decompositions, ranks them by estimated utility, and recursively refines these estimates to determine the best plans.

available database. This could be answered in several ways, each with different cost-accuracy trade-offs. For example, a single text snippet from Apple's 2023 annual 10-K forms provides a low-cost but highly summarized answer. Combining a snippet with a financial table yields more detail but requires additional table reasoning, increasing generation cost. Using multiple snippets from quarterly 10-Q filings offers a more comprehensive, fine-grained view, but incurs higher retrieval and generation costs. Finally, aggregating multiple tables across filings produces the most detailed analysis, yet requires both substantial retrieval and costly multi-table reasoning. These alternatives illustrate that solving the same task admits plans of varying granularity and expense, which in turn highlights the need for a planner that can dynamically adjust its reasoning path based on the available budget.

We refer to this capability as *optimizable planning*, the ability to systematically explore candidate plans and select the one that maximizes utility subject to a computational budget. Here, utility denotes the overall goodness of a plan under a given budget. In this view, planning is not only about producing a feasible reasoning trajectory but also about comparing alternatives under varying budget conditions.

We propose Optimizable LLM Planning, a *branch-and-bound* style framework for complex reasoning tasks. While branch-and-bound is a classic paradigm in optimization (Nelder et al., 1960), our work is the first to adapt it to planning with LLMs. In our setting, **branching** corresponds to enumerating multiple candidate plans by generating alternative decompositions of the task at each step. Each decomposition produces a distinct branch consisting of (i) an immediately solvable subtask that can be executed directly (e.g., a single retrieve-then-generate step in RAG), and (ii) a residual subtask that may require further reasoning and may depend on the result of the first. The **bounding** step is realized through *utility calibration*, which makes the abstract notion of plan "goodness" concrete by integrating both reward and cost into a measurable utility. The reward from the solvable subtask can be assessed directly through a task execution operator (e.g., retrieval, LLM reasoning, or agentic workflows) together with a reward model, while the residual subtask is evaluated indirectly by assigning lower and upper bounds on its utility. The lower bound is estimated by treating the residual in its current contextualized form as a task query, which typically yields a weaker (low-reward) signal since it has not yet been decontextualized. The upper bound reflects the best outcome the residual could achieve once fully specified. These bounds define an interval for the plan's utility, enabling principled ranking, early pruning of non-promising candidates, and efficient exploration without unrolling full trajectories. The process is applied *recursively*, where after solving the imme-

diate subtask, the residual is decontextualized based on its answer, *tightening* its utility bounds and *updating* the plan ranking. This refinement continues until the budget is exhausted or the top-ranked plan is found. The above process is illustrated in Figure 1.

Unlike greedy methods that commit to a single trajectory or heuristic tree search that explores broadly without budget awareness, OLP maintains a utility-driven branch-and-bound search that adapts continuously and efficiently as reasoning unfolds. While our framework still relies on heuristic models to calibrate reward and cost, the bounding mechanism ensures that these heuristics are used in a principled way: they approximate branch-and-bound by providing lower and upper utility estimates, which guide pruning and prioritization. Thus, although OLP does not guarantee global optimality, it systematically searches for the best plan among enumerated alternatives under the given budget.

We instantiated OLP on RAG tasks and evaluated it on both traditional multi-hop QA datasets as well as more recent complex agentic browsing datasets. OLP outperforms existing greedy search, beam search, and MCTS techniques, achieving superior accuracy with lower cost. Relative to simple strategies like greedy search, which prioritize efficiency over accuracy, OLP achieves a similar cost while improving accuracy by 2.04×. Compared to more advanced strategies such as beam search and MCTS, which favor accuracy over efficiency, OLP improves accuracy by 1.74× while reducing cost by 2.58×.

## 2 METHODOLOGY

Our framework addresses planning subject to a computational budget for complex tasks that require multi-step reasoning and interaction with external information sources. We formulate such tasks as a search problem over candidate plans, where each plan is a sequence of subtasks solved through actions (e.g., retrieve–then–generate) that collectively yield an answer while incurring computational cost. The objective is to identify the best plans that can be executed within the given budget. Achieving this requires exploring alternative plans rather than committing to a single one. However, exhaustively enumerating all execution plans is infeasible, and heuristic search methods offer no principled way to optimize planning under budget constraints.

To address this challenge, we formulate planning as a *plan ranking* problem, where candidate plans are ordered by their estimated *utility*. A plan's utility captures its likelihood of being the optimal choice within the given budget, allowing the planner to adapt its decisions as budget constraints change. Since computing the complete utility of each plan is impractical, OLP estimates it in a divide-and-conquer manner: each plan is decomposed into a simple subtask that can be solved directly and a residual subtask that requires further reasoning. The simple subtask provides immediate evidence of utility, while the utility of solving the residual subtask is approximated through bound estimation. The planner iteratively compares these utility estimates, refining them as more context becomes available, until it identifies a plan whose lower bound dominates all other upper bounds. This approach allows OLP to efficiently identify the best plan without fully executing every candidate, ensuring that planning remains optimizable under budget constraints.

The remainder of this section details the process of OLP: (1) divide-and-conquer plan enumeration describes *branching* of our framework, i.e., how the planner iteratively decomposes a task in a divide-and-conquer fashion, producing candidate plans; (2) utility modeling defines how plan utility is measured under a computational budget; (3) utility estimation and refinement, the *bounding* part of our framework, explains how utilities are approximated and iteratively tightened through decontextualization as planning progresses; and (4) plan pruning formalizes how candidate plans are compared and discarded based on their utilities under budget constraints. The full algorithm is described in Algorithm 1. In this section, we denote a set of possible plans as $T$, where each plan $\tau \in T$ is a sequence of subtasks: $\tau = [s_1, s_2, \ldots, s_n]$.

**Plan Enumeration via Divide-and-Conquer.** Our planning framework starts with generating multiple candidate decompositions of the input task, each of which represents an alternative plan for solving it. As shown in line 5 of Algorithm 1, the planning model produces several decompositions, where each plan consists of (i) an immediately solvable subtask and (ii) a residual subtask that may depend on the solution of the first. The immediately solvable subtask can be addressed directly (e.g., through a single generation or retrieve–then–generate step), while the residual subtask cap-

---

**Algorithm 1** Overview of our branch-and-bound framework that estimates the lower and upper bound utility of different candidate plans for efficient exploration.

---

**Require:** Task decomposer $\mathcal{DC}$, Task decontextualizer $\mathcal{DT}$, execution operator $\mathcal{O}$, utility model $\mathcal{U}$
1: **Input:** input task $t$, budget $B$, **Output:** top-$k$ plans with the highest utility
2: **Initialize:** the set of plans $T \leftarrow \{[t]\}$, the total cost $c \leftarrow 0$
3: **while** $c < B$ **do**
4:     **for** plan $\tau_i = [s_1, \ldots, s_{n-1}, s_n]$ in $T$ **do**
5:         Decompose the residual subtask $s_n$ into directly solvable subtask $s_n^1$ and residual subtask $s_n^2$ using $\mathcal{DC}$; Update $c$ with cost of decomposition
6:         Add plan $[s_1, \ldots, s_{n-1}, s_n^1, s_n^2]$ to $T$
7:         Compute the exact utility of $s_n^1$, and estimate the lower and upper bounds of the utility of $s_n^2$ using $\mathcal{U}$
8:         Prune plans with utility upper bound $<$ $k$-th largest utility lower bound of all plans or with cost lower bound exceeding the budget
9:     **if** there exists $k$ plans with utility lower bound $\geq$ utility upper bound of all other plans **then**
10:         Return these $k$ plans
11:     **for** plan $\tau_i$ in $T$ **do**
12:         Apply the execution operator $\mathcal{O}$ to $s_n^1$ to obtain artifact $\mathcal{O}(s_n^1)$; Update $c$ with cost of executing $\mathcal{O}$
13:         Decontextualize $s_n^2$ based on $s_n^1$ and $\mathcal{O}(s_n^1)$ using $\mathcal{DT}$; Update $c$ with cost of decontextualization
14:         Compute the exact utility of $s_n^2$ using $\mathcal{U}$
15: **Return** top plans

---

tures the remaining reasoning needed to complete the task. In this way, our framework enumerates a frontier of candidate plans by recursively decomposing the task in a divide-and-conquer fashion. Examples of decompositions that include solvable and residual subtasks are illustrated in Figure 1. Details of the prompts used by the planning model for this process are provided in Appendix A.

**Calibration of Utility.** In OLP, the utility of a plan $\tau$ is not directly observable and must instead be calibrated through measurable signals that approximate its quality under a computational budget. For each subtask $s \in \tau$, calibration proceeds in two stages. An execution operator $\mathcal{O}$ is first applied to $s$ under the current budget to produce an intermediate artifact $\mathcal{O}(s)$. This artifact is then evaluated by a reward model $\mathcal{R}$, which estimates how likely the subtask can be successfully solved given $\mathcal{O}(s)$, and a cost model $\mathcal{C}$, which estimates the resources consumed and thus the chance of exceeding the total budget $B$ in the future.

A key design is to aggregate reward and cost in a manner that reflects the compounding nature of multi-step plans. Because the failure or budget overconsumption of any individual subtask can compromise the entire plan, utility must penalize weak subtasks rather than allowing them to be masked by stronger ones. To this end, we define plan utility as a multiplicative combination of reward and cost terms across subtasks. This formulation jointly captures progress toward solving the task and adherence to budget constraints, while naturally enforcing that a plan's utility decreases if any subtask is either low-reward or highly costly.

Formally, the calibrated utility of a plan $\tau$ under budget $B$ is

$$u_\tau = \mathcal{U}(\tau, B) = \prod_{s \in \tau} u_s = \prod_{s \in \tau} \mathcal{R}\big(s, \mathcal{O}(s)\big)^w \cdot \mathcal{C}\big(\mathcal{O}(s), B\big)^{1-w} \tag{1}$$

where $u_s$ is the utility of subtask $s$ and $w \in [0, 1]$ balances the contributions of reward and cost.

In practice, we compute utility in log-space:

$$\log u_\tau = \sum_{s \in \tau} \Big[ w \log \mathcal{R}\big(s, \mathcal{O}(s)\big) + (1-w) \log \mathcal{C}\big(\mathcal{O}(s), B\big) \Big] \tag{2}$$

This representation yields a convex combination of $\log \mathcal{R}$ and $\log \mathcal{C}$, making the role of $w$ in mediating the trade-off explicit and mitigating numerical underflow. Through this calibration, utility provides a principled measure of how likely a plan is to achieve task success while remaining feasible under the computational budget.

This calibration process is generalizable to different settings. In the RAG setting, the operator $\mathcal{O}$ can be a search module that retrieves passages, the reward model $\mathcal{R}$ can be a reranker that scores their

relevance to the query, and the cost model $\mathcal{C}$ can be based on the number of retrieval and generation tokens, where using more tokens leads to a lower score. This choice provides a *lightweight* proxy for task progress without requiring OLP to generate a full answer at this stage.

More generally, for reasoning tasks beyond RAG, $\mathcal{O}$ could be a solver such as an LLM or an agentic workflow that produces a candidate derivation or answer. $\mathcal{R}$ can then score this output via LLM-as-a-judge or consistency checks, while $\mathcal{C}$ can evaluate cost based on FLOPs, latency, or energy predictors. By framing utility itself as a calibrated combination of reward and cost, OLP provides a unified and extensible mechanism for ranking and pruning candidate plans under budget constraints.

**Utility Estimation and Refinement.** As discussed above, computing the full utility of a plan $\tau$ is generally infeasible. Instead, OLP estimates utilities by assigning exact values to directly solvable subtasks and bounded estimates to residual subtasks. These bounds ($u^{lb}$ for lower bound and $u^{ub}$ for upper bound), capture the range of possible utility until additional information becomes available.

For a directly solvable subtask $s$ (line 7 in Algorithm 1), exact utility means lower and upper bounds are equal (i.e., $u_s = u_s^{lb} = u_s^{ub}$). This is estimated by combining the subtask reward with the cost of the LLM call that instantiates the subtask, either during plan enumeration or decontextualization.

For residual subtasks prior to decontextualization (line 7 in Algorithm 1), utility can only be approximated since they depend on upstream solvable subtasks. We therefore treat them as not self-contained: the reward estimated from the current contextualized form provides a conservative lower bound, while the upper bound is set to the maximum achievable reward. As an illustration, consider the RAG setting with the residual subtask "How do *they* compare to Apple's total quarterly net sales during 2023?" in Figure 1. Here, the reference *they* links back to upstream solvable subtasks, making the residual not self-contained. When we compute its utility with a reranker applied to documents retrieved by the search module, the reranker's score is taken as the lower bound. The upper bound is fixed to the maximum score the reranker can output (equal to 1).

As for cost, lower and upper bounds are estimated using the token budget of the forthcoming decontextualization call together with the minimum and maximum possible output tokens. These reward and cost estimates are then combined to produce utility bounds, which are progressively refined as planning unfolds.

Formally, the lower and upper bounds of a plan $\tau$ are computed as follows:

$$\log u_\tau^{lb} = \Big( \sum_{s_1,\ldots,s_{n-1}} \log u_{s_i} \Big) + \log u_{s_n} \; ; \; \log u_\tau^{ub} = \Big( \sum_{s_1,\ldots,s_{n-1}} \log u_{s_i} \Big) + \log u_{s_n}^{max} \qquad (3)$$

where $\tau = [s_1, \ldots, s_{n-1}, s_n]$, with each of the first $n-1$ subtasks directly solvable and the last subtask $s_n$ being residual, $u_{s_n}^{max}$ denotes its maximum achievable utility, while each $\log u_{s_i}$ is computed using Equation (2).

To refine these utility estimates, the planner performs decontextualization. As shown in line 13 of Algorithm 1, given a solvable subtask $s_i^1$ and its dependent residual subtask $s_i^2$, the decontextualizer obtains the solution to $s_i^1$ and rewrites $s_i^2$ so that it no longer depends on $s_i^1$ (prompt details in Appendix B). In line 14, the utility of the decontextualized residual is then recomputed with tighter bounds, reflecting the reduced uncertainty.

Because a decontextualized residual can itself be decomposed further, the planner recursively applies this decomposition–decontextualization cycle. In doing so, the utility bounds are progressively tightened and plan estimates become more accurate, enabling increasingly reliable ranking and pruning of candidate plans as the search proceeds.

**Pruning Plans.** Once utilities for the subtasks are estimated, the lower and upper bounds (denoted as $u_\tau^{lb}$ and $u_\tau^{ub}$) of an entire plan's utility can be computed via Equation (3). These bounds allow the planner to eliminate unpromising candidates without fully unrolling every plan. In particular, we check whether there exist $k$ plans such that

$$\min_{\tau_i \in \{\tau_1,\ldots,\tau_k\}} u_{\tau_i}^{lb} \geq \max_{\tau_j \notin \{\tau_1,\ldots,\tau_k\}} u_{\tau_j}^{ub} \qquad (4)$$

If this condition holds, we can safely conclude that the current top-$k$ plans (ranked by utility) have already been identified (lines 9 and 10 of Algorithm 1). Further unrolling of the remaining candidates cannot change the outcome, since their future utilities are bounded above by their current $u^{ub}$

values, while the selected plans already have $u^{lb}$ values that dominate these bounds (Fagin et al., 2001; Zhang et al., 2024).

Similarly, as described in line 8 of Algorithm 1, any plan $\tau_i$ can be pruned if

$$u^{ub}_{\tau_i} \; < \; \min_{\tau_j \in \{\tau_1, \ldots, \tau_k\}} u^{lb}_{\tau_j} \tag{5}$$

where $\tau_1, \ldots, \tau_k$ are the $k$ plans with the highest current $u^{lb}$. Such a plan is already dominated and cannot enter the top-$k$ set in the future. Pruning these dominated candidates improves efficiency by focusing exploration on plans whose bounds still leave room for optimality.

Finally, we also track the cost of producing the plan and its execution cost. If the cumulative cost of a plan $\tau$ exceeds the budget $B$, the plan is immediately discarded.

## 3 EXPERIMENTAL EVALUATION

A defining feature of OLP is its ability to optimize the likelihood of task success under varying budget constraints. Unlike existing search algorithms, it avoids explicit trajectory rollouts and prunes dominated plans, enabling more effective exploration. To evaluate these advantages, we study both accuracy and cost, asking two key questions:

- **No budget imposed:** How does OLP compare to existing methods when computation is unconstrained? This setting applies to domains (e.g., finance, medicine) where exhaustive reasoning is preferred and higher costs are acceptable in exchange for maximum result quality.
- **Budget-constrained**: How does OLP compare to existing methods across different budget levels? This setting reflects cost-sensitive domains (e.g., online services) where answers must remain accurate but reasoning must be adapted in granularity and efficiency to fit within the available budget.

We instantiate these evaluations on complex RAG tasks, which provide a representative proxy for real-world applications. Such tasks usually demand multi-step reasoning and careful decomposition, which makes them naturally amenable to being solved through alternative plans. In addition, RAG tasks involve interactions with external tools (e.g., retrievers, document corpora), a characteristic that closely mirrors real-world applications.

As outlined in Section 2, instantiating OLP requires specifying an execution operator $\mathcal{O}$, a reward model $\mathcal{R}$, and a cost model $\mathcal{C}$. For RAG tasks, we employ the dense retriever Snowflake-arctic-embed-m-v2.0 (Yu et al., 2024) as the operator, the reranker BGE-reranker-v2-minicpm-layerwise (Li et al., 2023; Chen et al., 2024) as the reward model, and a token-based pricing model as the cost model. The details of our experimental setup are provided in Section 3.1, and the corresponding results are reported in Section 3.2.

### 3.1 EXPERIMENTAL SETUP

**Datasets and metrics.** We evaluated our framework on two representative RAG datasets that require addressing complex user queries with document-based evidence: Musique (Trivedi et al., 2022), a multi-hop QA benchmark, and BrowseComp-Plus (Chen et al., 2025), a challenging dataset for deep research tasks. For accuracy, we reported exact match (EM) and F1 scores by comparing predicted answers against the gold references, as both datasets contain short-text answers. For cost, we adopted a token-based pricing model that accounts for all LLM calls during task execution, following OpenAI's pricing scheme.[2] To improve readability, we scale the cost by a factor of $10^3$.

**Baselines.** We evaluated our approach against several search algorithms built on the ReAct problem-solving framework, each representing a different accuracy–cost tradeoff. Broadly, we compare two types: methods that explore only a few trajectories, emphasizing cost reduction over accuracy, and methods that explore multiple trajectories, emphasizing accuracy at a higher cost.

---

[2]https://openai.com/api/pricing/

For the cost-focused case, we consider greedy search, which follows a single trajectory. Since ReAct by default uses chain-of-thought (CoT) reasoning at each step, we further include a baseline variant that removes CoT entirely, pushing cost minimization to the extreme.

For the accuracy-focused case, we consider beam search and Monte Carlo Tree Search (MCTS), both of which explore multiple actions per step to form multiple trajectories. Beam search retains the top-scoring partial plans at each step, while MCTS simulates rollouts to estimate rewards: an approach that is more computationally expensive but can achieve higher accuracy.

The details of these search algorithm implementations are provided below:

- Greedy: the default implementation of ReAct that operates iteratively, deciding at each step a *single* action (provide an answer or initiate a new reasoning step) through chain-of-thought, which may include interactions with external tools.

- Greedy without CoT: a greedy variant where the model outputs the action directly without chain-of-thought.

- Beam Search: Rather than pursuing a single action at each step, beam search maintains up to $k$ beams with the highest scores, thereby considering up to $k^2$ actions per step. The scores are computed using a reward model $\mathcal{R}$.

- MCTS: MCTS explores $k$ actions, retaining the best one at each step. It uses UCT to run multiple simulations, where actions are selected and rolled out to completion, producing scores (via a reward model $\mathcal{R}$) that inform subsequent UCT decisions. The action chosen most frequently is then selected.

To ensure fairness, all baseline methods employ the same BGE-reranker as our framework, using it either as the reward model or directly in the retrieval stage.

**Implementation details.** We adopted an open, off-the-shelf LLM, Qwen2.5-7B-Instruct, as the backbone for executing all methods. As noted earlier, we evaluate all methods under two execution settings: with and without a cost budget. When a budget is given, execution stops once the budget is consumed. Without a budget, since the methods run iteratively until producing an answer and may not converge, we cap the execution at 10 iterations to prevent indefinite runs, a limit chosen to be sufficiently large. In either setting, if the budget is consumed or the iteration cap is reached without an answer, the model is given a final opportunity to generate an answer to ensure task completion.

## 3.2 RESULTS

Table 1: Performance of all methods without a budget. **Bolded** and underlined numbers represent the performance of our method when it ranks as the best and second best, respectively.

|  | Musique | | | BrowseComp-Plus | | |
|---|---|---|---|---|---|---|
|  | EM | F1 | Cost ↓ | EM | F1 | Cost |
| Greedy w/o CoT | 12.0 | 18.6 | 1.12 | 11.0 | 13.8 | 6.47 |
| Greedy | 10.0 | 17.7 | 1.66 | 7.90 | 9.60 | 8.29 |
| Beam search | 13.3 | 22.7 | 4.38 | 11.4 | 14.3 | 29.85 |
| MCTS | 9.10 | 14.9 | 29.3 | 11.4 | 15.1 | 132.90 |
| OLP (ours) | **22.8** | **36.8** | **1.05** | **24.2** | **30.3** | **5.91** |

Table 1 and Table 2 report the performance of all methods without a budget and under varying budget levels, respectively. Across both settings, our method consistently outperforms the greedy baseline, whether or not chain-of-thought is used, while incurring comparable cost. This demonstrates that, given the same budget, OLP can explore multiple plans effectively and produce better answers.

When compared to more advanced search strategies (beam search and MCTS), we observe that although they generally improve over greedy, OLP still achieves higher accuracy at substantially lower cost. This advantage arises because our framework (1) avoids costly explicit rollouts by using efficient reward model estimates, and (2) prunes plans whose upper bounds are dominated by the lower bounds of others, thereby focusing exploration on the most promising candidates.

Table 2: Performance of all methods under different levels of budget.

| | Low budget | | | Mid budget | | | High budget | | |
|---|---|---|---|---|---|---|---|---|---|
| | EM | F1 | Relative cost ↓ | EM | F1 | Relative cost | EM | F1 | Relative cost |
| *Musique*: low budget = 0.5, mid budget = 1, high budget = 1.5 | | | | | | | | | |
| Greedy w/o CoT | 9.1 | 16.2 | 1.00x | 8.1 | 14.4 | 1.00x | 12 | 17.5 | 1.00x |
| Greedy | 11.0 | 15.7 | 1.13x | 14.4 | 19.7 | 1.30x | 11.1 | 15.8 | 1.36x |
| Beam search | 12.0 | 18.3 | 1.23x | 18.0 | 24.3 | 1.54x | 13.0 | 21.3 | 1.81x |
| MCTS | 10.0 | 18.6 | 1.28x | 14.3 | 23.9 | 1.68x | 11.2 | 20.4 | 2.02x |
| OLP (ours) | **13.4** | **24.8** | **0.75x** | **20.4** | **33.0** | 1.06x | **18.7** | **31.2** | 1.09x |
| *BrowseComp-Plus*: low budget = 2.5, mid budget = 5, high budget = 7.5 | | | | | | | | | |
| Greedy w/o CoT | 8.7 | 10.3 | 1.00x | 11.1 | 12.9 | 1.00x | 10 | 11.7 | 1.00x |
| Greedy | 6.4 | 10.6 | 1.15x | 6.5 | 8.6 | 1.23x | 5.5 | 9.0 | 1.32x |
| Beam search | 9.8 | 12.9 | 1.26x | 11.8 | 14.0 | 1.46x | 6.7 | 11.4 | 1.75x |
| MCTS | 10.8 | 13.9 | 1.28x | 7.7 | 10.7 | 1.64x | 10.6 | 12.0 | 2.07x |
| OLP (ours) | **18.1** | **22.8** | **0.78x** | **22.8** | **28.9** | 1.06x | **27.0** | **31.1** | 1.14x |

Furthermore, as shown in Table 2, the performance of OLP generally improves as the budget increases, highlighting its ability to leverage additional budget to yield more accurate answers.

## 4 ANALYSIS

Section 3 demonstrates the overall effectiveness of our approach compared to existing search algorithms. In this section, we present a deeper analysis to better understand the sources of this improvement (Section 4.1). We also evaluate how the performance of our framework varies with different reward models (Section 4.2).

### 4.1 UNDERSTANDING THE GAINS OF OUR FRAMEWORK

Section 3 demonstrates that OLP achieves higher result quality than existing search algorithms while maintaining cost comparable to simple greedy search, and substantially lower than more complex approaches such as beam search and MCTS. In this section, we provide a deeper analysis to understand why our framework delivers both quality and efficiency gains.

For quality, we measure whether OLP can explore more plans. As highlighted in Section 1, the key features of our framework are avoiding explicit rollouts of all candidate plans and pruning unpromising ones, which enable more effective budget allocation toward higher-quality plans. As a result, under the same budget or iteration limit, OLP can often explore more plans, leading to higher quality. The number of trajectories/ plans explored by each method is reported in Table 3.

For efficiency, we evaluate how many iterations each method requires to finish execution. Fewer iterations imply lower cost. We also measure the proportion of tasks that successfully produce an answer within the budget or iteration cap. As noted in Section 3.1, existing search algorithms lack convergence guarantees, so they may exhaust resources without yielding an answer, requiring extra computation to force a final output. Thus, we allow models to have a final opportunity to generate an answer, though it incurs additional cost. Results for these metrics are reported in Table 4.

From Table 3, we observe that OLP generally explores the most plans among all methods, naturally yielding higher quality. It also generally prunes 20–30% of dominated plans, demonstrating effective budget allocation toward higher-quality plans, which further boosts quality and efficiency.

As shown in Table 4, OLP also requires the fewest iterations to complete, resulting in greater efficiency and lower cost. In addition, it achieves the highest task completion rate, providing stronger empirical convergence guarantees while incurring minimal extra cost for generating answers.

### 4.2 IMPACT OF DIFFERENT REWARD MODELS ON PERFORMANCE

As outlined in Section 2, our framework leverages a reward model to estimate the rewards of subtasks, thereby signaling the likelihood of a plan's success. Consequently, the quality of these re-

Table 3: Total number of trajectories/plans explored by all methods, along with the proportion of plans pruned by our method, under both budgeted and non-budgeted settings.

| | Musique | | | | BrowseComp-Plus | | | |
|---|---|---|---|---|---|---|---|---|
| | None | Low | Mid | High | None | Low | Mid | High |
| Greedy w/o CoT | 1.00 | 1.00 | 1.00 | 1.00 | 1.00 | 1.00 | 1.00 | 1.00 |
| Greedy | 1.00 | 1.00 | 1.00 | 1.00 | 1.00 | 1.00 | 1.00 | 1.00 |
| Beam search | 5.64 | 1.92 | 2.59 | 3.20 | 6.18 | 3.07 | 3.37 | 4.10 |
| MCTS | 4.32 | 2.00 | 2.00 | 2.38 | 4.11 | 2.00 | 2.50 | 2.78 |
| OLP (ours) | 5.35 | **2.22** | **4.38** | **5.10** | **7.03** | 2.80 | **5.24** | **6.44** |
| % plans pruned in OLP | 28.6 | 4.50 | 21.7 | 27.1 | 33.1 | 9.6 | 30.2 | 30.9 |

Table 4: Number of execution iterations to complete and percentage of tasks with answer generated within the maximum allowed execution iterations.

| | Musique | | BrowseComp-Plus | |
|---|---|---|---|---|
| | #Iteration ↓ | %Task with answers | #Iteration | %Task with answers |
| Greedy w/o CoT | 4.08 | 85.0 | 4.08 | 84.6 |
| Greedy | 5.39 | 71.0 | 5.45 | 71.9 |
| Beam search | 3.85 | 85.7 | 4.74 | 76.1 |
| MCTS | 5.52 | 67.0 | 5.61 | 63.6 |
| OLP (ours) | **2.74** | **96.7** | **2.64** | **100.0** |

ward estimates is crucial. To assess this, we examine the performance of our method under reward models of varying strengths. In Section 3, we employed a 2.7B-parameter reranker (BGE-reranker-v2-minicpm-layerwise) as the reward model. For comparison, we also evaluate a smaller reranker, BGE-reranker-v2-m3, with 0.6B parameters.

Table 5 presents the results across different reward model sizes. Across both datasets, we find that smaller reward models, while somewhat less accurate, still maintain reasonable performance with costs remaining stable. Notably, even with the smaller reranker, our framework still surpasses the best search baseline that uses the 2.7B reranker, underscoring the effectiveness of our approach.

Table 5: Performance of our method under reward models of different sizes (and thus strength).

| | Musique | | | BrowseComp-Plus | | |
|---|---|---|---|---|---|---|
| | EM | F1 | Cost ↓ | EM | F1 | Cost |
| Best baseline + 2.7B reranker | 13.3 | 22.7 | 4.38 | 11.4 | 15.1 | 132.90 |
| OLP + 0.6B reranker | 20.4 | 29.5 | 0.92 | 15.6 | 20.5 | 5.70 |
| OLP + 2.7B reranker | 22.8 | 36.8 | 1.05 | 24.2 | 30.3 | 5.91 |

## 5  CONCLUSION

We introduce Optimizable LLM Planning (OLP), a branch-and-bound framework that formulates planning with LLMs as a budget-constrained optimization problem. Unlike greedy or heuristic search methods, OLP systematically enumerates and evaluates candidate plans through recursive decomposition, and calibrates their utility by integrating reward and cost. Bounding via utility intervals enables principled ranking, early pruning of dominated plans, and efficient exploration without unrolling entire trajectories. We instantiated OLP on retrieval-augmented generation (RAG) tasks across multi-hop QA and agentic browsing benchmarks. OLP consistently outperforms existing search strategies, matching greedy search in cost while improving accuracy by 2.04×, and surpassing beam search and MCTS with 1.74× higher accuracy at 2.58× lower cost.

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

## A  DECOMPOSITION PROMPT

Table 6: Prompts for decomposition.

| Decomposition prompt |
| --- |

You are an expert at question decomposition. Your task is to analyze and transform complex questions into simpler, more focused components.

Given a question, decompose it into two sub-questions by:

- Breaking down the complex question into two logical components
- The decomposition must be complete: combining the answers of sub-questions must be enough to solve the given question
- It is ok to have dependencies between the two sub-questions
- Providing multiple decomposition approaches when possible
- Do not attempt to solve the sub-questions

The output should be a JSON-formatted list where each element represents one decomposition approach:

```
[
  {
    "question 1": "First sub-question",
    "question 2": "Second sub-question"
  },
  {
    "question 1": "First sub-question",
    "question 2": "Second sub-question"
  }
]
```

Here is an example.

**Question:** How many academic staff are at the university in Budapest that has the official abbreviation BME ?

**Output:**

```
[
  {
    "question 1": "How many academic staff are at each
                   university in Budapest?",
    "question 2": "Which university from these universities
                   has the official abbreviation BME?"
  },
  {
    "question 1": "Which university in Budapest has the
                   official abbreviation BME?",
    "question 2": "How many academic staff are there at this
                   university?"
  },
  {
    "question 1": "What universities are in Budapest?",
    "question 2": "How many academic staff are there at the
                   university that has the official
                   abbreviation BME from these universities?"
  }
]
```

Your response must follow the output format without generating anything else.

# B    DECONTEXTUALIZATION PROMPT

Table 7: Prompts for decontextualization.

> **Decontextualization prompt**
>
> You are an expert at question rewriting. Your task is to analyze and transform complex questions into simpler, more focused components.
> Given question 1, question 2, and a document that can potentially be used to answer question 1, rewrite question 2 so that it is context-independent:
>
> - Remove any dependencies from question 1
> - Incorporate only the answer to question 1 based on the given document
> - Make question 2 self-contained and clear
>
> Here is an example.
> **Question 1:** What universities are in Budapest?
> **Question 2:** How many academic staff are there at the university that has the official abbreviation BME from these universities?
>
> **Document:**
> Document title: Budapest
> Document content: Budapest is home to several prestigious universities including the University of Veterinary Medicine, Corvinus University, Budapest University of Technology and Economics (BME), and Budapest University of Economics and Business.
>
> **Output:**
> ```
> Answer: University of Veterinary Medicine, Corvinus University,
> Budapest University of Technology and Economics (BME),
> and Budapest University of Economics and Business
>
> Question: How many academic staff are there at the university
> that has the official abbreviation BME among the University of
> Veterinary Medicine, Corvinus University, Budapest University
> of Technology and Economics (BME), and Budapest University
> of Economics and Business?
> ```
> Your response must follow the output format without generating anything else.

## C    THE USE OF LARGE LANGUAGE MODELS (LLMS)

LLM was used only to aid writing quality (proofreading and polishing grammar). No ideas, claims, methods, results, or references are generated by LLMs. All content decisions and revisions are made by the authors.

