# OpenReview forum: "Optimizable LLM Planning: A Branch-and-Bound Framework for Complex Tasks"
_ICLR.cc/2026/Conference — ICLR 2026 Conference Withdrawn Submission_

### Official Review · Reviewer_kRTW · 2025-10-29

**Soundness:** 2
**Presentation:** 1
**Contribution:** 1
**Rating:** 2
**Confidence:** 4

**Summary:**

This paper addresses the problem of LLM planning under a limited budget for solving complex tasks. Previous LLM planning approaches, e.g., CoT and tree-based search, are typically budget-blind and ad-hoc. To address this limitation, the authors propose an optimizable LLM planning framework under budget constraints called Optimizable LLM Planning (OLP). The framework consists of three processes: (1) plan branching via a divide-and-conquer scheme, (2) utility modeling to measure plan utility, (3) plan bounding through utility estimation, and (4) plan pruning based on estimated plan utility. To demonstrate its effectiveness and efficiency, the authors conduct experiments on two RAG tasks with various scenarios, achieving superior performance in both accuracy and efficiency.

**Strengths:**

1. **Clear Motivation.** The motivation and its explanation are well-articulated and easy to understand.
2. **Novel Contribution.** This is the first budget-aware optimized planning framework for LLMs.

**Weaknesses:**

1. **Limited Novelty in ML Community.** The approach primarily adapts classical algorithms to LLMs without introducing novel methodological contributions to the research community. The core algorithms rely on prompting, and the overall framework construction follows classic algorithmic principles. Even though there are some interesting and new ideas, such as branching approach, the level of novelty is not high, and it hasn't shown whether the current way is the best.
2. **Figure Quality Issues.** The figures require refinement as they are currently difficult to interpret:
    - Figure fonts are too small and hard to read.
    - The flow is not immediately clear, and the meaning of arrows is ambiguous.
    - The source of maximum achievable reward and minimum required cost is unclear, as is the derivation of lower and upper bounds. The explanations are too abstract to follow easily.
    - It is unclear how plans are ranked using the lower and upper bounds based on the figure alone.
3. **Presentation Could Be Improved.** The paper is text-heavy with limited use of formulations, algorithms, or pseudocode. There are also repeated lines, suggesting that the content and contribution to the community could be more substantial.
4. **Limited Evaluation Scope.** The proposed method and baselines are only tested on two RAG tasks. It remains unclear whether this approach generalizes to other domains. Additionally, while the figure mentions Math tasks, these are not included in the experimental setup.

**Questions:**

- Are you using a single LLM with different prompts, or multiple independent LLMs? This should be clarified in the algorithm description and formulation.
- In Table 1, for a fair comparison, shouldn't MCTS be given sufficient budget to achieve its optimal performance? Since MCTS typically requires more budget to perform well while OLP is designed to work efficiently with limited budget, it would be helpful to understand how much budget MCTS needs to match OLP's performance and how efficiently OLP performs when both methods are given ample budget. Additionally, it is not yet clear why OLP outperforms MCTS in the no-budget-limit setup. Please provide more detailed explanation of the evaluation protocols in the experimental setup.

**Details Of Ethics Concerns:**

An ethical statement and reproducibility statement are included. reproducibility statement

---

### Official Review · Reviewer_Kndd · 2025-10-30

**Soundness:** 2
**Presentation:** 2
**Contribution:** 3
**Rating:** 6
**Confidence:** 3

**Summary:**

The paper tackles the problem of planning with LLMs under an explicit computational budget for complex, multi-step tasks such as RAG-style multi-hop QA and agentic browsing. The authors argue that popular approaches like ReAct (greedy, stepwise reasoning) and tree-style searches (beam, MCTS) are essentially budget-blind — they cap computation with ad-hoc limits but do not optimize plan choice with respect to a budget. To address this, the paper proposes OLP, a branch-and-bound framework that (i) enumerates alternative task decompositions, (ii) assigns lower/upper utility bounds to each partial plan via a calibrated combination of reward and cost, and (iii) prunes dominated plans to focus exploration on candidates whose upper bounds still dominate. The key idea is to split each step into an immediately solvable subtask and a residual subtask, then decontextualize the residual once upstream subtasks are solved, so utility bounds can be tightened over time (Alg. 1). On RAG tasks (Musique, BrowseComp-Plus), OLP achieves substantially higher EM/F1 than greedy, beam search, and MCTS, while using less or comparable cost, e.g. on Musique OLP gets 22.8 EM / 36.8 F1 vs 13.3 / 22.7 for beam, at 1.05 vs 4.38 cost.

**Strengths:**

1. Divide-and-conquer decomposition with decontextualization: The “solvable subtask + residual subtask -> decontextualize residual -> tighten bounds” loop is elegant.
2. Strong experimental results: On Musique and BrowseComp-Plus, OLP dominates both **cost-oriented** (greedy) and **accuracy-oriented** (beam, MCTS) baselines, sometimes by large margins
3. Robustness to weaker reward models: Even with a smaller reranker (0.6B), OLP still beats the best baseline that uses the stronger reranker.

**Weaknesses:**

1. Presentation: Fig.1 is hard to fully understand the "bound" (see question 2).
2. Lack of related work.
3. Although Sec. 2 claims the framework is general to “execution operators” (LLM, retrieval, agentic workflows), all experiments are RAG-style QA/browsing. This narrows the evidence for claims about “complex tasks” and “LLM planning” in general.
4. The baselines are good (ReAct, beam, MCTS), but lack of comparison to other OLP methods in the experiments.

**Questions:**

1. In Eq. (1)–(2), how is the weight $\omega$ chosen in practice? Is it fixed?
2. How do you distinguish this work from [ReCode](https://www.arxiv.org/pdf/2510.23564)?
3. Could you elaborate on the meaning/mechanism of "bound" in the paper? Is it a bound that limits the exploration of the residual subtask?
4. Could you conduct experiments on other benchmarks that might also need to divide the plan into subplans, such as [ALFWorld](https://arxiv.org/abs/2010.03768) and [ScienceWorld](https://arxiv.org/abs/2203.07540)?
5. Could you discuss other OLP methods?

---

### Official Review · Reviewer_6y5e · 2025-10-31

**Soundness:** 1
**Presentation:** 1
**Contribution:** 2
**Rating:** 2
**Confidence:** 3

**Summary:**

The paper proposes OLP, a branch-and-bound planner for optimizable LLM planning in multi-step tasks. The method formulates planning as a budgeted optimization problem. At each step, the task is decomposed into a solvable subtask and a residual subtask, and the cost and reward are combined into a utility to discover the most promising plans. The method is evaluated on RAG problems that require reasoning, and it discovers more correct solutions than the baselines.

**Strengths:**

- The problem that the paper is tackling, budget-aware reasoning and planning in LLMs, is highly relevant.
- The solvable + residual split with decontextualization is elegant.
- Explicitly separating the executor operator, reward model, and cost model makes a lot of sense.
- Improving RAG performance in complex tasks is encouraging

**Weaknesses:**

- The method frames itself as a generic optimizable LLM planning paper. Figure 1 and its caption indicate that this method should be applicable to complex tasks such as math tasks, with an LLM as the execution operator. However, at the end, the method is only evaluated on RAG tasks. I would expect to see methods on, for instance, math benchmarks, coding benchmarks, and potentially even combinatorial planning benchmarks, to see if the method can work in settings where the reward model needs to be something other than a reranker.
- The budget is unclear. First off, does B correspond to the maximum overall cost of the planning process or to the maximum overall cost of the trajectory? To me, the former would be natural. However, in many places, it seems like B refers to the plan budget, not to the planning budget. For instance, L8 of Algorithm 1 states: "Prune plans ... with cost lower bound exceeding the budget". On L280, you write: "If the cumulative cost of a plan \tau exceeds the budget B, the plan is immediately discarded." If B were the planning budget, wouldn't it be exhausted before a single plan could exceed it? Furthermore, Table 1 indicates that OLP's cost is lower than Greedy without CoT, while Table 3 shows that OLP typically explores 2-7x more trajectories. This strongly suggests that the reported cost is the trajectory cost, not the planning cost! The cost and budget should be clearly defined in the paper.
- The ordering of the steps in Algorithm 1 is confusing. On L7, how can you compute the exact utility of s_n^1, before applying the execution operator O on L12? Eq. 1 indicates that the utility depends on the output O(s), which should be unavailable before L12. Similarly, how do you compute the exact utility of s_n^2 on L14?
- The poor performance of the MCTS baseline is very confusing. In Table 1 (Musique), the performance is worse than greedy, both without and with CoT! This would need a proper explanation. Essentially, you'd imagine MCTS to correspond to some sort of best-of-N algorithm, so not being able to outperform greedy is very surprising, and could indicate that the MCTS could be tuned to perform better. For instance, how are the MCTS actions selected? You'd need to use the same action creation prompt for MCTS as for the branch-and-bound for the comparison to be fair. The comparison could also benefit from using the reranker as a heuristic rather than relying on rollouts for value estimation. A budget-aware best-first search could also be a reasonable baseline to include.

**Questions:**

- I believe the paper would need much more transparent reporting of the results to understand the cost-benefit trade-off of this method. Preferably, the code should be released. In particular: What total number of input + output tokens is used by the underlying LLM, not just for the final plan but for the whole process? How many reranker calls? How many retriever calls?
- What is the branching factor, and how do you combat combinatorial explosion? Algorithm 1 indicates that at every iteration, you add steps into each plan, whereas Table 3 indicates that you typically prune around 20-30 % of the plans. For any branching factor, this doesn't seem like enough to combat exponential, uncontrollable growth?
- You write on L234: "the reward estimated from the current contextualized form provides a conservative lower bound." Why would this be a conservative lower bound? Isn't this more like the average, not a lower bound?
- How is the value of w chosen? How sensitive is the algorithm to that?
- There's a line of work for token-budget-aware LLM reasoning that seems relevant and is probably worth at least acknowledging [1, 2, 3]

Overall, while the idea is interesting and the RAG gains are encouraging, I think the paper is not yet ready for publication. The open questions regarding the methodology and budgeting should be clarified, and either the experimental section expanded or the paper reframed to better match the empirical validation.

[1] Han, T., Wang, Z., Fang, C., Zhao, S., Ma, S., & Chen, Z. (2024). Token-budget-aware llm reasoning. arXiv preprint arXiv:2412.18547.

[2] Li, J., Zhao, W., Zhang, Y., & Gan, C. (2025). Steering LLM Thinking with Budget Guidance. arXiv preprint arXiv:2506.13752.

[3] Wen, H., Wu, X., Sun, Y., Zhang, F., Chen, L., Wang, J., ... & Li, Y. (2025). Budgetthinker: Empowering budget-aware llm reasoning with control tokens. arXiv preprint arXiv:2508.17196.

---

### Official Review · Reviewer_pbpD · 2025-11-01

**Soundness:** 1
**Presentation:** 2
**Contribution:** 2
**Rating:** 2
**Confidence:** 4

**Summary:**

The proposed approach is a RAG technique that incorporates the ideas in the branch-and-bound search.
It wasn't clear how the proposed approach maintains the search tree from the branch-and-bound; the main idea around estimating the cost/utility and its usage for bounding is introduced in the paper.
The proposed approach was applied to two RAG-based benchmarks.

**Strengths:**

The proposed approach incorporates the use of heuristics to improve the search, which may be relevant for RAG-based problem solving with LLMs. In the experiment, the proposed approach shows improved performance compared with baselines.

**Weaknesses:**

Concept/terminology issue.
Inaccurate description of essential concepts.

No need to introduce "optimizable planning"
First of all, rational agents maximize the expected utility as mentioned.
Or, cost-optimal planning emphasizes the aspect of finding plans with the optimal cost because not all planning problems have the utility or cost, such as classical planning.

The branch-and-bound is not a paradigm for optimization. It is one of search frameworks that can be used for implementing optimization methods. Tree search can naturally handle budget costs.
Adapting branch-and-bound or similar ideas into LLM is common, under the extension of tree-of-thought style approaches.

Existing ideas applied to the LLM-based problem-solving technique with some inspiration and interpretations.
The main question is how that actually works.


Normally, branch-and-bound works by incorporating optimistic evaluation of the unsolved/remaining problems.
The proposed method cannot provide such an optimistic evaluation due to the nature of the methods, using LLMs.
It contradicts the term optimizable planning. It is a local search, and a less greedy approach at best.

Divide-and-conquer is a paradigm for polynomial-time problem solving.
Dynamic programming and Divide-and-conquer are different.
One cannot solve planning problems by Divide-and-conquer as the complexity class is beyond polynomial in general.

**Questions:**

Q1 In branch-and-bound, how does backtracking happen in the proposed approach?

Q2 In Eq. (2), how are the two models defined, the reward and cost model?
Is it within the scope of the paper or reusing the existing methods?

Q3: How do we know if a subtask is directly solvable and determine if it is solved?

Q4 Two RAG-based datasets were used in the experiment.
For the experiment result, could you compare it with the evaluation provided in the benchmark papers, such as BrowseComp-Plus?

---

### Note · Authors · 2025-12-04

I have read and agree with the venue's withdrawal policy on behalf of myself and my co-authors.